# Analyzing and forecasting under-5 mortality trends in Bangladesh using machine learning techniques

Shayla Naznin[1]*, Md Jamal Uddin[2,3], Ishmam Ahmad[4], Ahmad Kabir[2]

1 Department of Statistics, Mawlana Bhashani Science and Technology University, Tangail, Bangladesh,
2 Department of Statistics, Shahjalal University of Science and Technology, Sylhet, Bangladesh, 3 Faculty of Graduate Studies, Daffodil International University, Dhaka, Bangladesh, 4 FCPS (Internal Medicine) Part-II Trainee, Medicine Unit-2, Shaheed Suhrawardy Medical College Hospital, Dhaka, Bangladesh

* shayla.naznin@gmail.com

## Abstract

### Background

Under-5 mortality remains a critical social indicator of a country's development and economic sustainability, particularly in developing nations like Bangladesh. This study employs machine learning models, including Linear Regression, Ridge Regression, Lasso Regression, Bayesian Ridge, Decision Tree, Gradient Boosting, XGBoost, and CatBoost, to forecast future trends in under-5 mortality. By leveraging these models, the study aims to provide actionable insights for policymakers and health professionals to address persistent challenges.

### Methods

Data from the 1993–94 to 2017–18 Bangladesh Demographic and Health Survey (BDHS) was analyzed using advanced machine learning algorithms. Key metrics, including Mean Absolute Error (MAE), Root Mean Squared Error (RMSE), R-squared, and Mean Absolute Percentage Error (MAPE), were employed to evaluate model performance. Additionally, k-fold cross-validation was conducted to ensure robust model evaluation.

### Results

This study confirms a significant decline in under-5 mortality in Bangladesh over the study period, with machine learning models providing accurate predictions of future trends. Among the models, Linear Regression emerged as the most accurate, achieving the lowest MAE (4.05), RMSE (4.56), and MAPE (6.64%), along with the highest R-squared value (0.98). Projections indicate further reductions in under-5 mortality to 29.87 per 1,000 live births by 2030 and 26.21 by 2035.

### Conclusions

From 1994 to 2018, under-5 mortality in Bangladesh decreased by 76.72%. While the Linear Regression model demonstrated exceptional accuracy in forecasting trends,

**Data availability statement:** The primary data supporting the findings of this manuscript are derived from the publicly accessible BDHS (1993–94 to 2017–18) datasets, available through the Measure DHS website (https://dhsprogram.com/data/available-datasets.cfm; https://dhsprogram.com/pubs/pdf/PR104/PR104.pdf)

**Funding:** The author(s) received no specific funding for this work.

**Competing interests:** The authors declare that they have no competing interests.

long-term predictions should be interpreted cautiously due to inherent uncertainties in socio-economic conditions. The forecasted rates fall short of the Sustainable Development Goal (SDG) target of 25 deaths per 1,000 live births by 2030, underscoring the need for intensified interventions in healthcare access and maternal health to achieve this target.

## Introduction

Under-five mortality (U5M) is a critical indicator of child health and overall development in any country, and it remains a significant challenge for Bangladesh. The Sustainable Development Goals (SDGs), specifically Target 3.2, aim to reduce under-five mortality rates to 25 deaths per 1,000 live births by 2030 [1]. Despite significant progress under the Millennium Development Goals (MDGs), where Bangladesh successfully reduced U5M from 134 to 45 deaths per 1,000 live births between 1993 and 2018 [2], the country still faces considerable challenges. Bangladesh continues to have one of the highest U5M rates in South Asia, trailing behind Pakistan and India [3,4].

High Under-five mortality rates in Bangladesh stem from a range of socio-economic and healthcare-related factors [5–9]. Limited access to maternal healthcare services, high rates of illiteracy, early marriage, and adolescent fertility are significant contributors to the persistence of child mortality. As the country approaches the SDG deadline in 2030, accurately forecasting Under-five mortality trends becomes crucial to guide policy decisions and allocate resources effectively. However, there is a notable lack of studies projecting future U5M trends [10–13], leaving policymakers with limited evidence to address the remaining challenges in achieving the SDG targets.

Traditional statistical methods [14–16], though useful, often struggle to capture the complex, non-linear relationships among socio-economic, environmental, and healthcare-related factors influencing U5M [4,17–25]. In this context, machine learning (ML) techniques [26–30] offer a promising alternative. These methods excel in analyzing large datasets and identifying intricate patterns, which may be overlooked by conventional techniques. By leveraging ML algorithms, this study seeks to fill the gap in forecasting Under-five mortality trends in Bangladesh, providing insights to support evidence-based policymaking.

Bangladesh provides a unique context for this study as it has achieved remarkable progress in reducing under-5 mortality rates over recent decades, despite facing socio-economic and health system challenges. Insights gained from Bangladesh's trends and interventions can inform policies in other low- and middle-income countries with similar demographic and health profiles.

This research uses data from the Bangladesh Demographic and Health Survey (BDHS) spanning 1993–94 to 2017–18 to forecast Under-five mortality rates up to 2035. Through the application of machine learning techniques, it aims to project trends in Under-five mortality reduction and analyze the implications of these projections on achieving SDG targets. By addressing these gaps, the study contributes to the development of targeted interventions that can help Bangladesh overcome remaining challenges in reducing Under-five mortality. Ultimately, this work seeks to inform policymakers and health professionals, providing actionable insights for achieving sustainable progress in child health outcomes.

## Methods and materials

### Data

This study utilizes data from the Bangladesh Demographic and Health Surveys (BDHS) conducted between 1993–94 and 2017–18. The BDHS, implemented by the National Institute of Population Research and Training (NIPORT) in collaboration with ICF International,

provides nationally representative datasets capturing a comprehensive range of demographic and health indicators. These include under-5 mortality, fertility, maternal health, and child nutrition, among others.

The dataset focuses on under-5 mortality rates (U5MR), expressed as the number of deaths of children under five years of age per 1,000 live births. This indicator is crucial for assessing child health and survival over time, reflecting the broader socio-economic and public health environment in Bangladesh.

The data were acquired from the DHS Program's official portal following the necessary approval process. As the dataset is structured on a year-wise basis, no missing values were present. This eliminated the need for imputation or other missing data handling techniques. The dataset required minimal preprocessing, mainly involving standardization of variable definitions to ensure consistency across survey rounds. This approach facilitated a seamless comparison of Under-five mortality rate trends over the study period.

The BDHS datasets are publicly available upon request through the following link: https://dhsprogram.com/data/available-datasets.cfm.

## Target variable

The target variable in this study is the Under-5 Mortality Rate (U5MR), which represents the number of deaths of children under five years of age per 1,000 live births. The U5MR serves as a critical indicator of child health and survival reflecting the broader socio-economic and public health environment in Bangladesh over the studied period.

## Independent variable

We consider the year of the observation which is used as the primary temporal variable to analyze and forecast trends in under-5 mortality rates over time.

## Data preprocessing

Given the year-wise structure of the dataset, no missing values were present, eliminating the need for imputation or other missing data handling techniques. The dataset was small but comprehensive, covering under-5 mortality rates (U5MR) across several years. Preprocessing focused on ensuring consistency across survey rounds by standardizing variable definitions and aligning the data for analysis.

## Statistical analysis

This study employs a combination of traditional statistical approaches and machine learning (ML) models to analyze historical trends and project future trajectories of under-5 mortality in Bangladesh.

## Traditional Metrics

Two key metrics were used to quantify progress in reducing under-5 mortality over time:

1. **Overall Decline in Under-5 Mortality Rate (U5M):**

$$\text{Decline}\left(\%\right) = \left(\frac{U5M_t - U5M_0}{U5M_0}\right) \times 100$$

where $U5M_t$ is the under-5 mortality rate in year t, and
$U5M_0$ is the rate in the initial year.

2. **Annual Rate of Reduction (ARR):**

$$r = \left( \sqrt[n]{\frac{U5M_t}{U5M_0}} - 1 \right) \times 100,$$

where n represents the number of years between two survey rounds.

**Machine learning models.** A variety of ML models were used to explore linear and non-linear patterns in the data, including:

**Linear Models:** Linear Regression, Ridge Regression, Lasso Regression, and Bayesian Ridge.

**Tree-Based Models:** Decision Tree, Gradient Boosting, XGBoost, and CatBoost.

These models were chosen for their ability to capture both straightforward and complex relationships in time-series data. Tree-based models like CatBoost and XGBoost were particularly useful for handling non-linearities and feature interactions, while linear models provided insights into the overall trend due to the largely linear nature of U5MR over time.

**Visualization of data.** To visually understand the patterns in the dataset and assess the performance of various models, the under-5 mortality rate (U5MR) was plotted against predicted and forecasted values across different machine learning models are in Fig 1. These visualizations highlight how well each model captures both historical trends and future projections, with linear models aligning closely to observed data due to the dataset's linearity, while tree-based models addressed potential non-linearity's

**Model training and evaluation.** The models were trained using historical U5MR data from 1993 to 2018 and validated through time-based cross-validation. Forecasts for the period 2020–2035 were generated. The following evaluation metrics were employed to assess model performance:

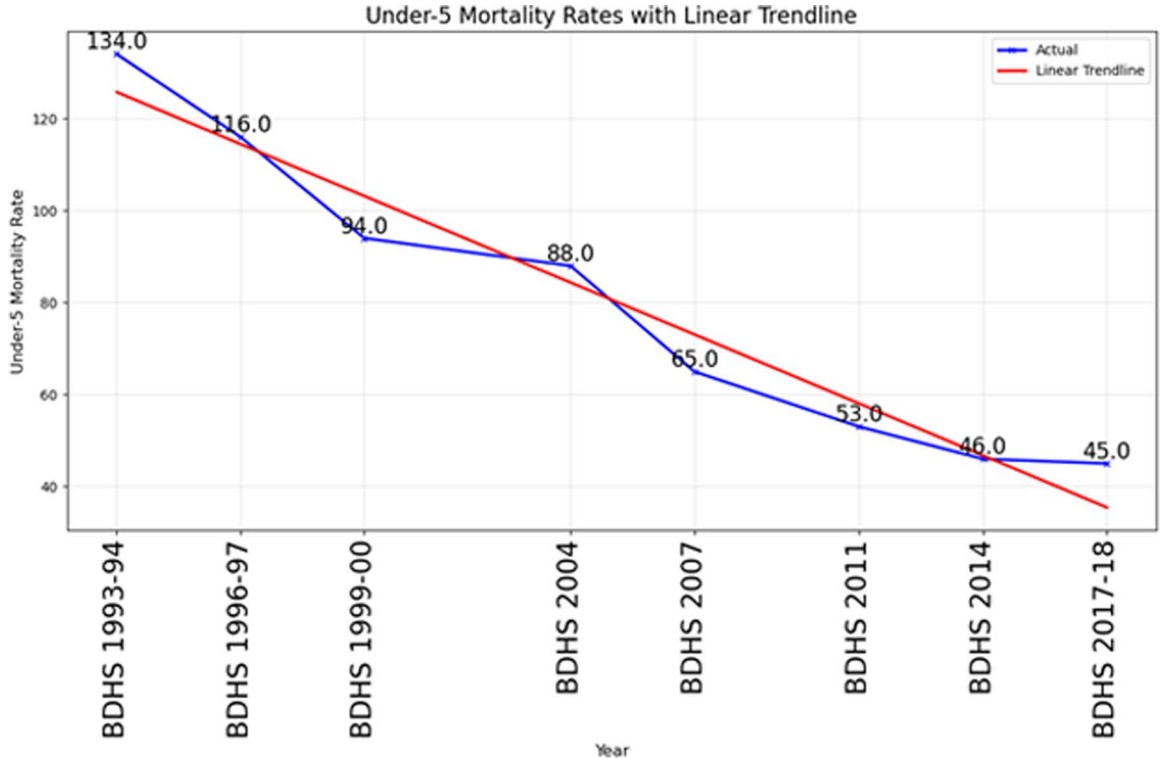

**Fig 1. Visualization of under-5 mortality rate in Bangladesh.**

**Mean Absolute Error (MAE):** Measures the average error in absolute terms.
**Root Mean Squared Error (RMSE):** Penalizes larger errors more heavily than MAE.
**R-squared ($R^2$):** Indicates how well the model explains the variability in U5MR.
**Mean Absolute Percentage Error (MAPE):** Evaluates the relative prediction error as a percentage.

The selection of multiple machine learning models was driven by the need to comprehensively capture both linear and non-linear trends in U5MR. Linear models were included because the data exhibited a largely linear decline over time, allowing these models to perform well. On the other hand, non-linear models like CatBoost and XGBoost were included to account for potential complexities in the data, such as fluctuations or interactions with other latent variables. This mixed-methods approach of statistical analysis and machine learning provided robust insights into U5MR trends. The forecasts generated offer critical information for policymakers, enabling targeted interventions to further reduce under five mortality in Bangladesh.

**Ethics approval and consent to participate.** Permission to use the data was obtained from the Measure DHS program through formal legal registration. The study utilized BDHS (1993–94 to 2017–18) datasets, which are publicly available through the Measure DHS website (https://dhsprogram.com/data/available-datasets.cfm).

## Results

### Model performance comparison

The results of the model comparison reveal important insights into the performance of different machine learning algorithms for predicting under-five mortality trends. Table 1 shows each model was evaluated using key metrics such as Mean Absolute Error (MAE), Root Mean Squared Error (RMSE), R-squared and Mean Absolute Percentage Error (MAPE).

The Linear Regression model exhibits excellent performance in forecasting under-5 mortality rates with an MAE of 4.05 and an RMSE of 4.56, indicating minimal errors in predictions. Its R-squared value of 0.98 suggests that the model explains 98% of the variance in the data, demonstrating its strong predictive capability. The MAPE of 6.64% further confirms the model's accuracy reflecting a low average percentage error. Overall, Linear Regression is highly effective and reliable for predicting under-5 mortality trends.

Similar to Linear Regression, the Ridge Regression model also performs exceptionally well, with an MAE of 4.12 and an RMSE of 4.61. The R-squared value remains high at 0.98, indicating that the model captures 98% of the variance in under-5 mortality rates. The MAPE of 6.73% is slightly higher than Linear Regression but still indicates strong predictive accuracy.

**Table 1. Model performance comparison.**

| Models | MAE (Mean Absolute Error) | RMSE (Root Mean Squared Error) | R-squared | MAPE (Mean Absolute Percentage Error) |
|---|---|---|---|---|
| Linear Regression | 4.05 | 4.56 | 0.98 | 6.64 |
| Ridge Regression | 4.12 | 4.61 | 0.98 | 6.73 |
| Lasso Regression | 4.06 | 4.57 | 0.98 | 6.66 |
| Bayesian Ridge | 4.37 | 4.81 | 0.98 | 7.03 |
| Decision Tree | 12.50 | 13.66 | 0.81 | 14.36 |
| Gradient Boosting | 12.50 | 13.66 | 0.81 | 14.36 |
| XGBoost | 15.00 | 15.30 | 0.76 | 19.08 |
| CatBoost | 10.55 | 11.55 | 0.87 | 12.09 |

Lasso Regression also shows impressive performance, closely mirroring the results of Linear and Ridge Regression. With an MAE of 4.06 and an RMSE of 4.57, the model demonstrates low error rates. The R-squared value of 0.98 signifies that it effectively explains 98% of the variance in the data. The MAPE of 6.66% reflects a minor average percentage error, underscoring the model's accuracy.

The Bayesian Ridge model performs well, though with slightly higher errors compared to the other linear models. It has an MAE of 4.37 and an RMSE of 4.81, indicating accurate but marginally less precise predictions. The R-squared value of 0.98 remains strong, showing that the model explains 98% of the variance in under-5 mortality rates. However, the MAPE of 7.03% suggests a slightly higher average percentage error, indicating room for improvement.

The Decision Tree model exhibits significantly higher errors compared to the linear models, with an MAE of 12.50 and an RMSE of 13.66. The R-squared value of 0.81 indicates that the model explains only 81% of the variance in the data, which is considerably lower. The MAPE of 14.36% reflects a higher average percentage error, indicating less reliable predictions. While Decision Tree models are useful for capturing non-linear relationships, in this context, they appear less effective for forecasting under-5 mortality trends.

The Gradient Boosting model mirrors the performance of the Decision Tree, with identical MAE and RMSE values of 12.50 and 13.66, respectively. The R-squared value of 0.81 suggests that it explains a similar portion of the variance in the data. However, like the Decision Tree, the MAPE of 14.36% indicates a higher percentage error, making it less suitable for accurate forecasting.

XGBoost performs poorly in this forecasting task, with the highest error rates among all models evaluated. The MAE of 15.00 and RMSE of 15.30 highlight significant inaccuracies in predictions. The R-squared value of 0.76 indicates that it explains only 76% of the variance, the lowest among the models tested. The MAPE of 19.08% further underscores the model's limitations, reflecting a high average percentage error.

The CatBoost model shows moderate performance with an MAE of 10.55 and an RMSE of 11.55. Its R-squared value of 0.87 suggests that it explains 87% of the variance in under-5 mortality rates which is lower than the linear models but higher than XGBoost and Decision Tree. The MAPE of 12.09% indicates a relatively higher average percentage error suggesting that while CatBoost is more accurate than some models it still falls short compared to linear models.

The limited performance of non-linear models may stem from the dataset's characteristics, where non-linear relationships and complex interactions are less pronounced. This highlights the importance of understanding the underlying data patterns when selecting machine learning models. While the models demonstrated strong forecasting accuracy based on internal validation (e.g., cross-validation), the study lacks external validation to confirm their applicability to unseen data. Future research could address this by testing the models on newer datasets or datasets from different regions, ensuring their robustness and generalizability. This limitation should be considered when interpreting the results.

Overall, linear models emerged as the most reliable for predicting under-5 mortality trends in Bangladesh, effectively capturing the predominant patterns with minimal error. Non-linear models, while potentially powerful for datasets with complex interactions, underperformed due to the linear nature of the trends in U5MR. These findings underscore the importance of aligning model selection with data characteristics to achieve robust predictions.

Table 2 presents the cross-validation accuracy results for different models across various fold splits. Each model was evaluated using 2-fold, 3-fold, 5-fold and 7-fold cross-validation techniques. The k-fold cross-validation results for different models provide insights into how each model performs under varying numbers of folds, revealing the consistency and stability of their predictions.

**Table 2. K-fold cross validation accuracy of different models.**

| Model | MAE | | | | RMSE | | | |
|---|---|---|---|---|---|---|---|---|
| | **2-Fold** | **3-Fold** | **5-Fold** | **7-Fold** | **2-Fold** | **3-Fold** | **5-Fold** | **7-Fold** |
| Linear Regression | 8.2727 | 8.8281 | 6.5825 | 8.5038 | 9.7256 | 9.5675 | 6.7063 | 8.5766 |
| Ridge Regression | 8.3422 | 8.8690 | 6.6111 | 8.5184 | 9.8014 | 9.5962 | 6.7274 | 8.5890 |
| Lasso Regression | 8.2808 | 8.8357 | 6.5877 | 8.5071 | 9.7344 | 9.5734 | 6.7103 | 8.5795 |
| Bayesian Ridge | 8.4698 | 9.0569 | 6.7178 | 8.6115 | 9.9452 | 9.7571 | 6.8195 | 8.6740 |
| Decision Tree | 18.2500 | 17.7778 | 14.9000 | 15.2143 | 22.4518 | 20.6831 | 15.7819 | 15.3795 |
| Gradient Boosting | 18.2499 | 17.7780 | 14.8999 | 15.2143 | 22.4516 | 20.6831 | 15.7818 | 15.3795 |
| XGBoost | 22.8753 | 15.5003 | 12.0001 | 13.1431 | 27.4518 | 17.3856 | 12.7101 | 13.1856 |
| LightGBM | 18.2260 | 18.1607 | 14.7982 | 15.3047 | 22.3444 | 20.8653 | 15.6119 | 15.4477 |
| CatBoost | 16.7438 | 15.4039 | 10.7930 | 11.7671 | 20.3535 | 17.7882 | 11.5839 | 11.8072 |
| Model | R-squared | | | | MAPE | | | |
| | **2-Fold** | **3-Fold** | **5-Fold** | **7-Fold** | **2-Fold** | **3-Fold** | **5-Fold** | **7-Fold** |
| Linear Regression | 0.8541 | 0.8251 | 0.7466 | 0.9790 | 10.8034 | 11.2994 | 9.4540 | 12.1875 |
| Ridge Regression | 0.8531 | 0.8253 | 0.7474 | 0.9785 | 10.8389 | 11.3319 | 9.4785 | 12.1883 |
| Lasso Regression | 0.8540 | 0.8251 | 0.7467 | 0.9789 | 10.8073 | 11.3065 | 9.4592 | 12.1885 |
| Bayesian Ridge | 0.8509 | 0.8212 | 0.7477 | 0.9767 | 10.9102 | 11.5813 | 9.6369 | 12.3008 |
| Decision Tree | 0.4732 | 0.3066 | 1.0463 | 0.8120 | 20.1377 | 20.5628 | 18.5876 | 18.5943 |
| Gradient Boosting | 0.4732 | 0.3066 | 1.0463 | 0.8121 | 20.1377 | 20.5632 | 18.5877 | 18.5946 |
| XGBoost | 1.1562 | 0.4842 | 1.0304 | 0.7642 | 27.8231 | 18.3918 | 15.6676 | 16.5089 |
| LightGBM | 0.4123 | 0.3022 | 1.0232 | 0.8655 | 20.2115 | 21.2345 | 18.6415 | 18.8663 |
| CatBoost | 0.4723 | 0.5337 | 0.6726 | 0.9840 | 18.6146 | 17.3421 | 13.2305 | 13.9504 |

Linear Regression shows strong and consistent performance across different fold values, with the 5-fold cross-validation producing the best results. The MAE (6.58) and RMSE (6.71) are the lowest in the 5-fold validation, indicating that this model's predictions are most accurate and consistent under this configuration. The R-squared values range from 0.75 to 0.98 with the highest performance in the 7-fold suggesting that Linear Regression explains the variance in the data well especially when more folds are used. The MAPE varies slightly with the lowest error observed in the 5-fold validation at 9.45% indicating a strong predictive ability.

Ridge Regression exhibits similar patterns to Linear Regression, with the best performance in the 5-fold cross-validation (MAE: 6.61, RMSE: 6.73). The R-squared values range from 0.75 to 0.98 indicating a strong fit across different folds. The MAPE is lowest in the 5-fold validation at 9.48% reflecting accurate predictions. Ridge Regression like Linear Regression, performs consistently across different k-folds with the 5-fold validation being the most optimal.

Lasso Regression also performs well across all folds with its best results in the 5-fold cross-validation (MAE: 6.59, RMSE: 6.71). The R-squared values show similar patterns to the other linear models with the highest value (0.98) in the 7-fold. The MAPE is lowest in the 5-fold validation at 9.46%, indicating that Lasso Regression effectively balances predictive accuracy with feature selection particularly in the 5-fold configuration.

Bayesian Ridge performs well though slightly less consistently than the other linear models. The best performance is observed in the 5-fold cross-validation (MAE: 6.72, RMSE: 6.82). R-squared values are high across all folds with a peak of 0.98 in the 7-fold validation. The MAPE is slightly higher than the other linear models with the lowest error at 9.64% in the 5-fold validation. Bayesian Ridge is reliable but exhibits slightly more variability in performance across different folds.

The Decision Tree model shows substantial variability across different folds, with the best performance in the 5-fold validation (MAE: 14.90, RMSE: 15.78). However, the model's R-squared values are significantly lower than the linear models ranging from 0.31 to 1.05. The MAPE is also high, with the lowest error at 18.59% in the 5-fold validation, indicating that the Decision Tree model is less effective and more prone to overfitting or underfitting particularly in smaller fold values.

Gradient Boosting mirrors the performance of the Decision Tree as expected with identical MAE, RMSE and R-squared values across all folds. The model performs best in the 5-fold validation with MAE and RMSE values of 14.90 and 15.78 respectively. However, like the Decision Tree Gradient Boosting shows lower R-squared values and high MAPE (18.59% in the 5-fold) indicating that while the model captures complex patterns it struggles with consistency and accuracy across different folds.

XGBoost shows considerable variability in performance across the different folds with the best results in the 5-fold validation (MAE: 12.00, RMSE: 12.71). However, the R-squared values are inconsistent, peaking at 1.03 in the 5-fold but dropping significantly in other folds. The MAPE is also relatively high with the lowest error at 15.67% in the 5-fold validation. XGBoost, while powerful, seems less stable across varying fold values, indicating a potential need for further tuning.

LightGBM exhibits similar variability to XGBoost, with its best performance in the 5-fold validation (MAE: 14.80, RMSE: 15.61). R-squared values vary widely, peaking at 1.02 in the 5-fold validation and the MAPE is high across all folds with the lowest error at 18.64%. LightGBM appears less consistent and less accurate than the linear models particularly when it comes to predicting under-5 mortality rates.

CatBoost shows moderate performance with the best results in the 5-fold validation (MAE: 10.79, RMSE: 11.58). The R-squared value peaks at 0.98 in the 7-fold validation indicating good explanatory power. The MAPE is lowest in the 5-fold validation at 13.23% suggesting that CatBoost can be an effective model though it is still outperformed by the linear models in this context.

Overall, the linear models (Linear, Ridge and Lasso Regression) consistently outperform the other models across different fold values, particularly in the 5-fold cross-validation, where they exhibit the lowest errors and highest R-squared values. Decision Tree, Gradient Boosting, XGBoost, LightGBM and CatBoost show more variability and higher errors indicating that they are less suited for this specific forecasting task when evaluated through k-fold cross-validation.

Based on the k-fold cross-validation results across different folds (2-fold, 3-fold, 5-fold and 7-fold), Linear Regression is the best-performing model for forecasting under-5 mortality trends in Bangladesh. Linear Regression consistently shows the lowest Mean Absolute Error (MAE) and Root Mean Squared Error (RMSE) across most folds, especially in the 5-fold validation where it achieved a MAE of 6.58 and RMSE of 6.71. The model consistently demonstrates high R-squared values particularly in the 7-fold validation (0.98) indicating that it explains the variance in the data very well. The Mean Absolute Percentage Error (MAPE) is also low across the folds with the lowest error in the 5-fold validation at 9.45% highlighting its predictive accuracy. Linear Regression provides the most reliable and accurate predictions for this dataset making it the best choice among the models evaluated. So, Regression stand out as the most reliable models for forecasting under-5 mortality trends in Bangladesh.

## Trends in and projection of under-5 mortality rate

Examining the BDHS data from 1993–94 to 2017–18 provides a comprehensive view of the progress in reducing under-5 mortality rates in Bangladesh. The Table 3 shows a significant

**Table 3.  Under-5 mortality rate and projections in Bangladesh.**

| Survey Name | Under 5 Mortality rate (per 1,000 live births) | Annual Rate of Reduction (ARR) (%) |
|---|---|---|
| BDHS 1993–94 | 134 | – |
| BDHS 1996–97 | 116 | −4.69 |
| BDHS 1999–00 | 94 | −6.77 |
| BDHS 2004 | 88 | −1.64 |
| BDHS 2007 | 65 | −9.61 |
| BDHS 2011 | 53 | −4.97 |
| BDHS 2014 | 46 | −4.61 |
| BDHS 2017–18 | 45 | −0.55 |
| Annual Rate of Reduction (ARR) 1993 to 2018 | −4.44 | |
| Change (%) during 1993–2018 | −76.72 | |
| Projection for the year 2020 | 37.19 | |
| Projection for the year 2025 | 33.53 | |
| Projection for the year 2030 | 29.87 | |
| Projection for the year 2035 | 26.21 | |

decline in the under-5 mortality rate (U5MR) which fell from 134 deaths per 1,000 live births in 1993–94 to 45 per 1,000 live births by 2017–18. This represents a 76.72% reduction over this period highlighting the substantial impact of health improvements increased access to healthcare and enhanced socio-economic conditions. The annual rate of reduction (ARR) during this timeframe was 4. 44% indicating a steady and consistent decrease in mortality rates.

The Annual Rate of Reduction (ARR) in under-5 mortality varied significantly across survey periods, reflecting changes in the effectiveness of interventions. From 1993 to 2000, U5MR saw a rapid decline, with the ARR peaking at -6.77% between 1996–97 and 1999–00. However, from 2004 to 2011, the ARR slowed to −4.61% to −1.64%, indicating challenges in sustaining rapid progress. Notably, between 2014 and 2017–18, the ARR stagnated at −0.55%, highlighting the need for renewed focus on interventions to further reduce child mortality.

Looking ahead projections suggest that this positive trend will continue. Based on current data and forecasting models the U5MR is expected to further decrease to 37.19 per 1,000 live births by 2020. This downward trend is anticipated to persist, with the rate projected to drop to 33.53 by 2025, 29.87 by 2030 and 26.21 by 2035. These projections reflect ongoing improvements in healthcare systems and public health interventions which are likely to continue reducing child mortality rates in the future. The use of machine learning models in this analysis offers valuable insights into these trends and supports the development of targeted strategies to further enhance child health outcomes in Bangladesh.

## Discussion

The primary objective of this study is to analyze to the under-five mortality trends in Bangladesh from 1993 to 2018 and project the future trends. A clear downward trend is observed in the analysis as shown in the Fig 2 highlighting a significant reduction in under-five mortality rates in Bangladesh over the years.

This study demonstrates a steady decline in Under-5 Mortality rates in Bangladesh over the past 20 years (1994–2018) is 76.72% aligning with southern Asian trends [4,31]. The annual reduction rate (ARR) for Under-5 Mortality rates in Bangladesh was 4.44%, which closely mirrors the Southern Asian ARR during the same period [4,31]. This study applied various machine learning models to forecast under-5 mortality trends in Bangladesh using

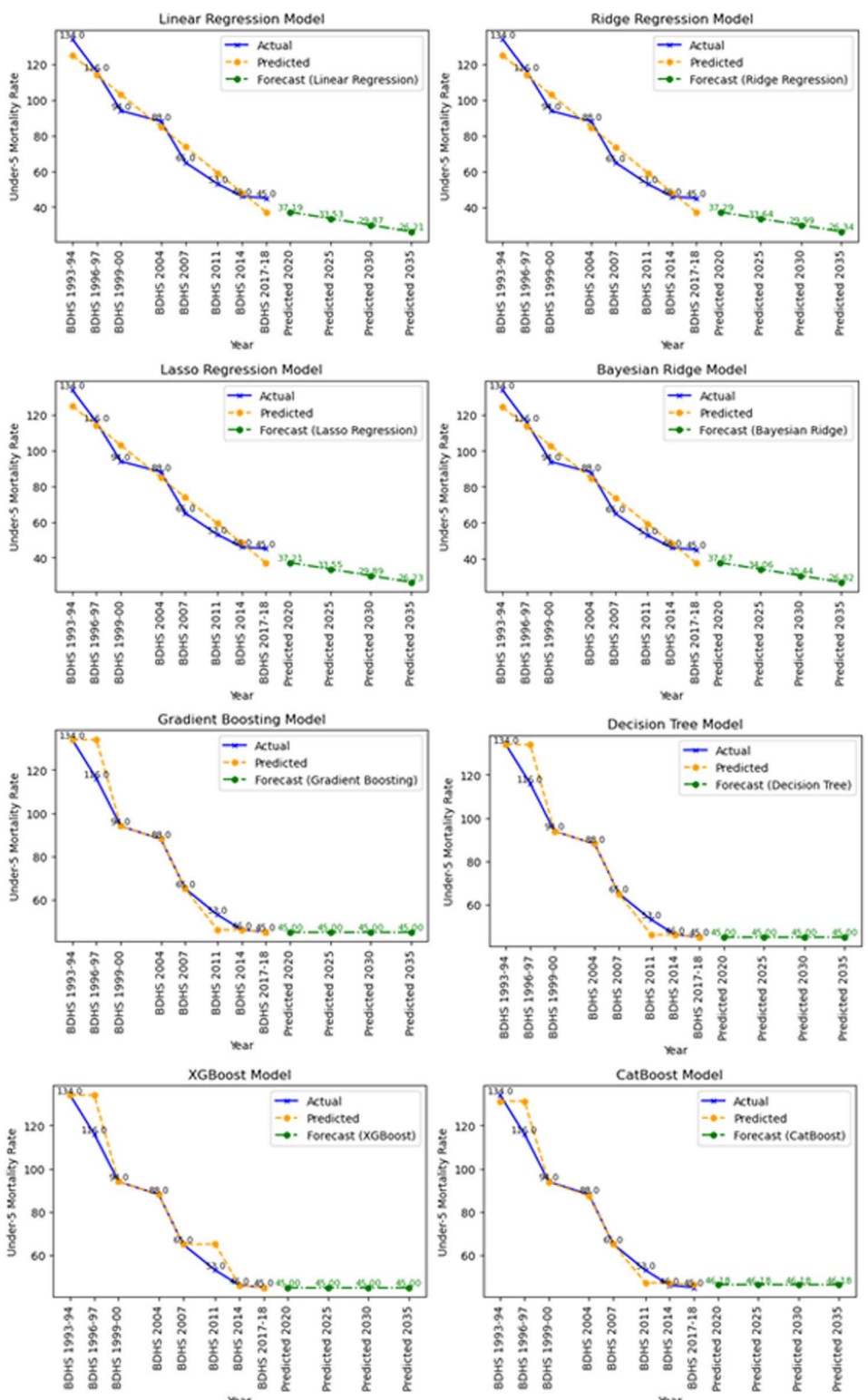

**Fig 2. Trends and projection of under-5 mortality in Bangladesh.**

data from the BDHS 1993–94 to 2017–18 surveys. The findings reveal a significant decline in under-5 mortality by 76.72% over the study period, indicating substantial progress in child health outcomes. However, projections suggest that Bangladesh may not meet the Sustainable Development Goal (SDG) target of reducing under-5 mortality to 25 deaths per 1,000 live births by 2030, as the forecasted rates are 29.87 by 2030 and 26.21 by 2035. The Linear Regression model emerged as the most effective forecasting tool among the machine learning models tested, achieving the lowest MAE (4.05), RMSE (4.56), MAPE (6.64%), and the highest R-squared value (0.98). This aligns with previous research suggesting that simpler models can outperform complex ones in time series data with clear linear trends [31]. The strong performance of the Linear Regression model indicates a stable and predictable relationship between time and under-5 mortality rates in Bangladesh.

Other machine learning models like Gradient Boosting, XGBoost, and CatBoost did not perform as well in this context. These models are typically advantageous in handling nonlinear relationships and large datasets with multiple features [32]. The relatively small dataset and the linear nature of the trend in under-5 mortality may have limited the effectiveness of these complex models. The significant reduction in under-5 mortality aligns with national and international reports highlighting Bangladesh's progress in child health [33]. Factors contributing to this decline include improved maternal education, increased access to healthcare services, vaccination programs, and socioeconomic development [34]. Despite these advancements, the projected shortfall in meeting the SDG target underscores the need for enhanced policy interventions focused on the most vulnerable populations.

To bridge the gap between the projected rates and the SDG target, policymakers should prioritize interventions that address the underlying causes of under-5 mortality. This includes improving nutrition, expanding immunization coverage, enhancing maternal health services, and addressing socioeconomic disparities [35]. Targeted strategies in rural and underserved areas are crucial, as disparities in healthcare access continue to affect child mortality rates [36]. Furthermore, leveraging real-time forecasting applications and web-based platforms can enhance the utility of machine learning models by providing dynamic and accessible predictions. These tools can facilitate timely policy adjustments, improve resource allocation, and allow health officials to respond swiftly to emerging trends or crises.

The use of machine learning models in this study highlights their potential in public health forecasting [37–41]. Accurate predictions of health indicators can inform policy decisions and resource allocation. However, the selection of appropriate models is critical as demonstrated, simpler models may suffice when dealing with linear trends and limited data while more complex models may be necessary for datasets with nonlinear patterns and multiple influencing factors [42]. The integration of web-based dashboards and mobile applications can make these models more interactive and actionable, enabling stakeholders to monitor progress and evaluate interventions in real time.

Future research should consider incorporating additional variables that influence under-5 mortality such as maternal health indicators, environmental factors and healthcare accessibility. Expanding the dataset and including more predictors may improve the forecasting accuracy of complex machine learning models. Moreover, continual monitoring and updating of models with recent data will enhance the reliability of predictions and support timely policy interventions.

## Strengths and limitations

This study's strengths include the utilization of a comprehensive time series dataset from the BDHS spanning 1993–94 to 2017–18, providing a robust foundation for analyzing long-term trends in under-5 mortality (U5M) in Bangladesh. The use of machine learning models, including Random Forest, Linear Regression, Ridge Regression, Lasso Regression, Bayesian

Ridge, Decision Tree, Gradient Boosting, XGBoost, LightGBM, CatBoost, allows for a nuanced approach to forecasting U5M trends, capturing both linear and non-linear patterns in the data. This multifaceted model application provides valuable insights into the effectiveness of different algorithms for public health forecasting.

However, the study has notable limitations. The machine learning models may not fully account for key socioeconomic and environmental factors influencing U5M, potentially limiting the accuracy and robustness of the forecasts. Additionally, the dataset relies on self-reported, retrospective data from household surveys, which may introduce recall bias, under-reporting, or over reporting of critical variables. These biases could affect the reliability of the mortality estimates and, consequently, the accuracy of the projections. The potential for such data inaccuracies should be considered when interpreting the results.

Also we state that this study's long-term forecasts rely on historical trends, assuming consistent influencing factors, but unexpected changes in socio-economic conditions or healthcare policies could impact accuracy. Forecasting models are sensitive to data variability, especially over extended horizons. Future studies should incorporate scenario-based modeling and updates with new data to enhance reliability.

Furthermore, the limited number of time points in the dataset could restrict the generalizability of the models, particularly for forecasting long-term trends. The retrospective and cross-sectional nature of the data further hinders the ability to establish causal relationships between mortality outcomes and underlying determinants. Addressing these limitations in future studies by incorporating real-time data, broader covariates, and longitudinal datasets could improve the accuracy of forecasts and strengthen the causal inferences drawn from the findings.

## Conclusion

This study underscores the utility of machine learning models in forecasting under-5 mortality (U5M) trends in Bangladesh, providing valuable data-driven insights to inform public health strategies. By leveraging data from the BDHS 1993–94 to 2017–18, models like Linear Regression, Ridge Regression, Lasso Regression, and CatBoost have demonstrated strong predictive capabilities, with each model excelling in capturing different dimensions of the data. The analysis projects a promising reduction in U5M to 26.42 per 1,000 live births by 2030, reflecting Bangladesh's significant progress in improving child health outcomes over the past decades.

However, these projections must be interpreted with caution, as they may be overly optimistic if socio-economic conditions, healthcare accessibility, or policy interventions change unfavorably. The models do not fully account for potential future disruptions, such as economic downturns, environmental crises, or emerging health challenges, which could hinder progress toward the SDG target of reducing U5M to 25 per 1,000 live births by 2030.

To bridge the gap between forecasted values and the SDG target, policymakers should focus on targeted interventions that address persistent challenges, such as disparities in healthcare access, undernutrition, and maternal health services. Resources should be prioritized for rural and underserved regions, where child mortality rates remain disproportionately high. Additionally, expanding vaccination coverage and improving socioeconomic conditions can accelerate progress toward achieving the SDG target.

Future research should aim to refine these forecasting models by incorporating complex social and environmental variables, such as climate change impacts, healthcare infrastructure, and maternal education levels. Utilizing real-time data and longitudinal datasets can enhance the models' ability to predict and adapt to emerging trends. Furthermore, integrating machine learning-based tools into web-based or mobile platforms could allow for dynamic, real-time forecasting, improving accessibility and facilitating timely policy interventions.

In conclusion, this study highlights the transformative potential of machine learning in public health forecasting and underscores its role in guiding evidence-based strategies to reduce child mortality. However, achieving the SDG target will require sustained efforts, innovative policy solutions, and continual improvements in forecasting methodologies to ensure their robustness and applicability to evolving public health challenges.

## Supporting information

**S1 Data. Under 5 mortality rate (per 1,000 live births).**
(DOCX)

## Author contributions

**Conceptualization:** Md Jamal Uddin, Ahmad Kabir.

**Data curation:** Shayla Naznin, Ishmam Ahmad.

**Formal analysis:** Shayla Naznin.

**Methodology:** Shayla Naznin, Ishmam Ahmad.

**Supervision:** Md Jamal Uddin, Ahmad Kabir.

**Validation:** Shayla Naznin.

**Visualization:** Shayla Naznin.

**Writing – original draft:** Shayla Naznin.

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
