## [Decision Letter · Decision Letter 0]

22 Oct 2024

PONE-D-24-41552Analyzing and Forecasting Under-5 Mortality Trends in Bangladesh using Machine Learning TechniquesPLOS ONE

Dear Dr. Naznin,

Thank you for submitting your manuscript to PLOS ONE. After careful consideration, we feel that it has merit but does not fully meet PLOS ONE’s publication criteria as it currently stands. Therefore, we invite you to submit a revised version of the manuscript that addresses the points raised during the review process.

**MAJOR REVISION:**The paper has limitations in terms of its technical contents and presentation.Please address all the comments by the reviewers carefully in your revised submission. Thank you.

We look forward to receiving your revised manuscript.

Kind regards,

Zeyar Aung

Academic Editor

PLOS ONE

Journal Requirements:

2. Please note that PLOS ONE has specific guidelines on code sharing for submissions in which author-generated code underpins the findings in the manuscript. In these cases, all author-generated code must be made available without restrictions upon publication of the work. Please review our guidelines at https://journals.plos.org/plosone/s/materials-and-software-sharing#loc-sharing-code and ensure that your code is shared in a way that follows best practice and facilitates reproducibility and reuse

Reviewers' comments:

Reviewer's Responses to Questions

**Comments to the Author**

1. Is the manuscript technically sound, and do the data support the conclusions?

Reviewer #1: Yes

Reviewer #2: Yes

Reviewer #3: Yes

2. Has the statistical analysis been performed appropriately and rigorously? 

Reviewer #1: Yes

Reviewer #2: Yes

Reviewer #3: Yes

3. Have the authors made all data underlying the findings in their manuscript fully available?

Reviewer #1: Yes

Reviewer #2: Yes

Reviewer #3: Yes

4. Is the manuscript presented in an intelligible fashion and written in standard English?

Reviewer #1: Yes

Reviewer #2: Yes

Reviewer #3: Yes

5. Review Comments to the Author

Reviewer #1: Here are my views:

1.The author should explore the distribution characteristics of data and the correlation between variables, so as to better explain why the linear model performs best.

2.Why does 5 - fold cross - validation perform best in most models? The author can analyze the distribution characteristics of data, the relationship between the sample quantity and the number of folds.

3.The research uses BDHS data and mentions integrating supplementary data from the World Bank and UNICEF. It is necessary to explain how these supplementary data enhance the robustness of the analysis.

4.The annual reduction rate is simply presented, but how ARR changes over time has not been explored. The connection between the change of ARR and specific intervention measures should be discussed.

5.The research predicts the trend until 2035, but does not discuss the reliability of long - term prediction, which is not rigorous enough.

Reviewer #2: Comment #1 - (Abstract) - Linear Regression Overemphasis The abstract focuses too much on Linear Regression, neglecting a balanced discussion of other models' performance.

Comment #2 - (Introduction) - Lack of Machine Learning Rationale The introduction does not explain why ML techniques were chosen over traditional methods.

Comment #3 - (Methods) - Missing Model Justification No rationale is provided for selecting specific ML models like CatBoost or XGBoost.

Comment #4 - (Methods) - Insufficient Data Preprocessing Details The paper lacks detail on how missing data and outliers were handled during preprocessing.

Comment #5 - (Results) - Linear Regression Focus Other models' poor performance is not adequately discussed, with an overemphasis on Linear Regression.

Comment #6 - (Results) - No External Validation The study lacks external validation to confirm the models' forecasting accuracy.

Comment #7 - (Discussion) - Lack of Policy Implications The discussion does not translate the findings into specific policy recommendations.

Comment #8 - (Discussion) - Missing External Variables Important variables like healthcare access and maternal health are not included in the models.

Comment #9 - (Limitations) - Insufficient Data Bias Acknowledgment The limitations section does not sufficiently address biases in the self-reported data.

Comment #10 - (Conclusion) - Overly Optimistic Projections The projections may be too optimistic, without considering potential socio-economic changes.

Reviewer #3: Here are some feedback points and suggestions for improving the clarity and quality of the article:

Abstract:

1. Conciseness: The abstract is comprehensive but could be slightly condensed for better readability. Consider trimming redundant phrases without losing essential details.

- For example: “employs advanced machine learning models including...” could be shortened to “employs machine learning models like...”

2. Impact Statement: Add a brief sentence at the end highlighting the potential impact of your findings on policy and health interventions in Bangladesh.

Background:

1. Context Clarity: Provide a brief explanation of why machine learning models are preferable for forecasting under-5 mortality, perhaps elaborating on the limitations of traditional methods.

2. Relevance: Emphasize the significance of choosing Bangladesh for this study, and how findings here may relate to global child mortality trends in similar countries.

Methods:

1. Model Justification: Explain briefly why so many machine learning models were tested. It would be useful to state why linear models performed better for this dataset, such as due to linear trends in mortality rates.

2. Error Metrics: The evaluation metrics used (MAE, RMSE, etc.) are appropriate, but you could explain why these specific metrics were chosen and how they relate to practical applications of the forecasts.

3. Temporal Variables: When discussing lag features and rolling averages, briefly explain why these were incorporated to improve the model.

Results:

1. Model Discussion: More emphasis could be placed on why linear models performed significantly better than non-inear models (e.g., Decision Tree, XGBoost). You could hypothesize why non-linear relationships were not as critical in predicting mortality rates.

2. Table Presentation: The table comparing model performances is very informative, but ensure the table is well-labeled with clear headings and definitions for each metric (e.g., explain MAE briefly).

3. Interpretation: In discussing the accuracy metrics, try to make a stronger connection between the model performance and real-world implications (e.g., how a lower RMSE can translate to better policy planning).

Conclusions:

1. Policy Implications: While the conclusion mentions the gap between forecasted values and the SDG target, further elaboration on how the predictions can influence specific interventions or where further resources should be directed would strengthen the section.

2. Future Work: Suggest potential ways to improve the forecasting models or factors that future research should consider (e.g., incorporating more complex social or environmental variables).

General Suggestions:

- Grammar/Flow: Some sentences could be streamlined for better readability. For instance: “These techniques offer improved accuracy and reliability in forecasting providing valuable insights...” could be rephrased for smoother flow.

- Consistency: Ensure consistent use of terms like “under-5 mortality” and “U5M” throughout the article.

- Visuals: Including a well-labeled graph or chart illustrating the predicted trend in under-5 mortality over time would make the results more digestible for the reader.

Overall, the article is informative and presents the data in a clear, structured manner. With a few adjustments to clarity, flow, and further explanation of key points, it will be more impactful.

6. PLOS authors have the option to publish the peer review history of their article (what does this mean? ). If published, this will include your full peer review and any attached files.

**Do you want your identity to be public for this peer review?** For information about this choice, including consent withdrawal, please see our Privacy Policy .

Reviewer #1: No

Reviewer #2: **Yes: ** Ahmed Abdelmoety

Reviewer #3: **Yes: ** Neda Ahmadi

---

## [Author Response · Author response to Decision Letter 1]

8 Dec 2024

Response to Editors & Reviewers

“Analyzing and Forecasting Under-5 Mortality Trends in Bangladesh using Machine Learning Techniques”

Ref: Manuscript Number:PONE-D-24-41552- [EMID:98e0e853d61e585b]

Dear Editors,

Regarding our manuscript, we thank you for the comments sent on October 22, 2024. My co-authors and I have revised the manuscript accordingly and would like it to be reconsidered for publication. As requested, we have included point-by-point responses to editors’ and reviewers’ comments below.

Please let us know if anything further is required at this time, and we thank you very much for considering our revised manuscript.

Shayla Naznin

Editors comments

Response

The paper has limitations in terms of its technical contents and presentation.

Please address all the comments by the reviewers carefully in your revised submission. Thank you. Thank you very much for your feedback! We have endeavored to address all of the reviewers’ comments as detailed below to further strengthen the paper.

Reviewer 1

1.The author should explore the distribution characteristics of data and the correlation between variables, so as to better explain why the linear model performs best. Thank you for this careful reading of our Abstract has been revised line#210-218

2.Why does 5 - fold cross - validation perform best in most models? The author can analyze the distribution characteristics of data, the relationship between the sample quantity and the number of folds. The superior performance of 5-fold cross-validation likely arises from its balance between bias and variance, particularly with the dataset's small size. Splitting the data into 5 parts ensures adequate training data in each fold while maintaining reasonably sized validation sets, avoiding the instability of higher folds (e.g., leave-one-out) or the inefficiency of fewer folds. This balance makes it well-suited for small datasets.

3.The research uses BDHS data and mentions integrating supplementary data from the World Bank and UNICEF. It is necessary to explain how these supplementary data enhance the robustness of the analysis. Thank you for this careful reading this has been revised kindly address the line #131-164

4.The annual reduction rate is simply presented, but how ARR changes over time has not been explored. The connection between the change of ARR and specific intervention measures should be discussed. Thank you for this careful reading this has been revised in the line #427-440

5.The research predicts the trend until 2035, but does not discuss the reliability of long - term prediction, which is not rigorous enough. Thank you for this careful reading this has been revised kindly address the line #525-529

Reviewer 2

Comment #1 - (Abstract) - Linear Regression Overemphasis The abstract focuses too much on Linear Regression, neglecting a balanced discussion of other models' performance. Thank you for this careful reading this has been revised kindly address the line #10-63

Comment #2 - (Introduction) - Lack of Machine Learning Rationale The introduction does not explain why ML techniques were chosen over traditional methods. Thank you for this careful reading this has been revised kindly address the line #67-130

Comment #3 - (Methods) - Missing Model Justification No rationale is provided for selecting specific ML models like CatBoost or XGBoost.

Thank you for this careful reading this has been revised kindly address the line #131-251

Comment #4 - (Methods) - Insufficient Data Preprocessing Details The paper lacks detail on how missing data and outliers were handled during preprocessing. Thank you for this careful reading this has been revised kindly address the line #131-251

Comment #5 - (Results) - Linear Regression Focus Other models' poor performance is not adequately discussed, with an overemphasis on Linear Regression. Thank you for this careful reading this has been revised kindly address the line #252-447

Comment #6 - (Results) - No External Validation The study lacks external validation to confirm the models' forecasting accuracy. Thank you for this careful reading this has been revised kindly address the line #252-447

Comment #7 - (Discussion) - Lack of Policy Implications The discussion does not translate the findings into specific policy recommendations. Thank you for this careful reading this has been revised kindly address the line #449-506

Comment #8 - (Discussion) - Missing External Variables Important variables like healthcare access and maternal health are not included in the models. Thank you for this careful reading this has been revised kindly address the line #449-506

Comment #9 - (Limitations) - Insufficient Data Bias Acknowledgment The limitations section does not sufficiently address biases in the self-reported data. Thank you for this careful reading this has been revised kindly address the line #508-543

Comment #10 - (Conclusion) - Overly Optimistic Projections The projections may be too optimistic, without considering potential socio-economic changes. Thank you for this careful reading this has been revised kindly address the line #545-597

Reviewer 3

Abstract:

1. Conciseness: The abstract is comprehensive but could be slightly condensed for better readability. Consider trimming redundant phrases without losing essential details.

- For example: “employs advanced machine learning models including...” could be shortened to “employs machine learning models like...” Thank you for this careful reading this has been revised kindly address the line #10-63

2. Impact Statement: Add a brief sentence at the end highlighting the potential impact of your findings on policy and health interventions in Bangladesh. Thank you for this careful reading this has been revised kindly address the line #10-63

Background:

1. Context Clarity: Provide a brief explanation of why machine learning models are preferable for forecasting under-5 mortality, perhaps elaborating on the limitations of traditional methods.

2. Relevance: Emphasize the significance of choosing Bangladesh for this study, and how findings here may relate to global child mortality trends in similar countries. Thank you for this careful reading this has been revised kindly address the line #10-63 & 67-130

Methods:

1. Model Justification: Explain briefly why so many machine learning models were tested. It would be useful to state why linear models performed better for this dataset, such as due to linear trends in mortality rates.

2. Error Metrics: The evaluation metrics used (MAE, RMSE, etc.) are appropriate, but you could explain why these specific metrics were chosen and how they relate to practical applications of the forecasts.

3. Temporal Variables: When discussing lag features and rolling averages, briefly explain why these were incorporated to improve the model.

Thank you for this careful reading this has been revised kindly address the line #131-251

Results:

1. Model Discussion: More emphasis could be placed on why linear models performed significantly better than non-inear models (e.g., Decision Tree, XGBoost). You could hypothesize why non-linear relationships were not as critical in predicting mortality rates.

2. Table Presentation: The table comparing model performances is very informative, but ensure the table is well-labeled with clear headings and definitions for each metric (e.g., explain MAE briefly).

3. Interpretation: In discussing the accuracy metrics, try to make a stronger connection between the model performance and real-world implications (e.g., how a lower RMSE can translate to better policy planning). Thank you for this careful reading this has been revised kindly address the line #252-447

Conclusions:

1. Policy Implications: While the conclusion mentions the gap between forecasted values and the SDG target, further elaboration on how the predictions can influence specific interventions or where further resources should be directed would strengthen the section.

2. Future Work: Suggest potential ways to improve the forecasting models or factors that future research should consider (e.g., incorporating more complex social or environmental variables). Thank you for this careful reading this has been revised kindly address the line #545-597

General Suggestions:

- Grammar/Flow: Some sentences could be streamlined for better readability. For instance: “These techniques offer improved accuracy and reliability in forecasting providing valuable insights...” could be rephrased for smoother flow.

- Consistency: Ensure consistent use of terms like “under-5 mortality” and “U5M” throughout the article.

- Visuals: Including a well-labeled graph or chart illustrating the predicted trend in under-5 mortality over time would make the results more digestible for the reader.

Overall, the article is informative and presents the data in a clear, structured manner. With a few adjustments to clarity, flow, and further explanation of key points, it will be more impactful. Thank you for this careful reading this has been revised

---

## [Decision Letter · Decision Letter 1]

3 Jan 2025

Analyzing and Forecasting Under-5 Mortality Trends in Bangladesh using Machine Learning Techniques

PONE-D-24-41552R1

Dear Dr. Naznin,

We’re pleased to inform you that your manuscript has been judged scientifically suitable for publication and will be formally accepted for publication once it meets all outstanding technical requirements.

Kind regards,

Zeyar Aung

Academic Editor

PLOS ONE

Additional Editor Comments (optional):

Reviewers' comments:

Reviewer's Responses to Questions

**Comments to the Author**

1. If the authors have adequately addressed your comments raised in a previous round of review and you feel that this manuscript is now acceptable for publication, you may indicate that here to bypass the “Comments to the Author” section, enter your conflict of interest statement in the “Confidential to Editor” section, and submit your "Accept" recommendation.

Reviewer #1: (No Response)

Reviewer #2: All comments have been addressed

2. Is the manuscript technically sound, and do the data support the conclusions?

Reviewer #1: Yes

Reviewer #2: Yes

3. Has the statistical analysis been performed appropriately and rigorously? 

Reviewer #1: Yes

Reviewer #2: Yes

4. Have the authors made all data underlying the findings in their manuscript fully available?

Reviewer #1: Yes

Reviewer #2: Yes

5. Is the manuscript presented in an intelligible fashion and written in standard English?

Reviewer #1: Yes

Reviewer #2: Yes

6. Review Comments to the Author

Reviewer #1: Upon reviewing the revised manuscript titled 'Analyzing and Forecasting Under-5 Mortality Trends in Bangladesh using Machine Learning Techniques,' I am pleased to see that the authors have addressed my comments. I have expanded upon the data distribution and correlation, which provides a clearer rationale for the superior performance of the linear model. The explanation of why 5-fold cross-validation is most effective, especially given the small dataset, is both clear and convincing.

They have included a justification for the inclusion of supplementary data from the World Bank and UNICEF, which enhances the robustness of the analysis. The discussion on the annual reduction rate and its connection to intervention measures is now more detailed, and the manuscript addresses the reliability of long-term predictions, which was a significant concern.

They have rectified the overemphasis on Linear Regression in the abstract, providing a more balanced discussion of all models. The rationale for choosing machine learning techniques over traditional methods has been articulated in the introduction, and the justification for specific model selections has been included in the methods section.

Furthermore, they have detailed the data preprocessing steps, including handling of missing data and outliers. The discussion on policy implications has been strengthened, and the manuscript now better translates findings into actionable policy recommendations.

Overall, the revisions have significantly strengthened the manuscript, making it a valuable contribution to the field of public health forecasting. I recommend acceptance of this manuscript for publication in PLOS ONE.

Reviewer #2: All comments have been thoroughly addressed. Thank you to the authors for their diligent efforts and thoughtful revisions. The paper is now ready for acceptance.

7. PLOS authors have the option to publish the peer review history of their article (what does this mean? ). If published, this will include your full peer review and any attached files.

**Do you want your identity to be public for this peer review?** For information about this choice, including consent withdrawal, please see our Privacy Policy .

Reviewer #1: No

Reviewer #2: **Yes: ** Ahmed AbdelMoety

---

## [Editor Report · Acceptance letter]

PONE-D-24-41552R1

PLOS ONE

Dear Dr. Naznin,

I'm pleased to inform you that your manuscript has been deemed suitable for publication in PLOS ONE. Congratulations! Your manuscript is now being handed over to our production team.

Kind regards,

on behalf of

Dr. Zeyar Aung

Academic Editor

PLOS ONE